# Physical Function of Japanese Preadolescents during the COVID-19 Pandemic

**DOI:** 10.3390/healthcare10122553

**Published:** 2022-12-16

**Authors:** Sho Narahara, Tadashi Ito, Yuji Ito, Hideshi Sugiura, Koji Noritake, Nobuhiko Ochi

**Affiliations:** 1Department of Pediatrics, Aichi Prefecture Mikawa Aoitori Medical and Rehabilitation Center for Developmental Disabilities, Okazaki 444-0002, Japan; 2Three-Dimensional Motion Analysis Laboratory, Aichi Prefectural Mikawa Aoitori Medical and Rehabilitation Center for Developmental Disabilities, Okazaki 444-0002, Japan; 3Department of Integrated Health Sciences, Graduate School of Medicine, Nagoya University, Nagoya 461-8673, Japan; 4Department of Pediatrics, Graduate School of Medicine, Nagoya University, Nagoya 466-8550, Japan; 5Department of Orthopedic Surgery, Aichi Prefectural Mikawa Aoitori Medical and Rehabilitation Center for Developmental Disabilities, Okazaki 444-0002, Japan

**Keywords:** child, COVID-19, exercise test, muscle strength, postural balance

## Abstract

Children’s exercise habits have changed during the COVID-19 pandemic. This study aimed to examine the physical function and physical activity of preadolescent children before and during the COVID-19 pandemic. This cross-sectional study compared time spent in moderate-to-vigorous physical activity (MVPA), grip strength, single-leg standing time, and two-step tests of healthy children aged 10 to 12 years, enrolled from January 2018 to January 2020 (pre-COVID-19 group, *n* = 177) and from January 2021 to September 2022 (during-COVID-19 group, *n* = 69). The during-COVID-19 group had weaker grip strength (median: 14.4 vs. 15.8 kg; *p* = 0.012), worse performance on the two-step test (mean: 1.56 vs. 1.60; *p* = 0.013), and less MVPA (median: 4 vs. 7 h per week; *p* = 0.004). Logistic regression showed that the during-COVID-19 group was significantly related to weaker grip strength (odds ratio: 0.904, 95% CI: 0.829–0.986; *p* = 0.022) and worse performance in the two-step test (odds ratio: 0.976, 95% CI: 0.955–0.997; *p* = 0.028). The COVID-19 pandemic decreased exercise opportunities for preadolescent children, which may have had a negative impact on muscle strength and balance. It is essential to increase the amount of MVPA among preadolescent children.

## 1. Introduction

In Japan, the first COVID-19 case occurred in January 2020, and a state of emergency was declared in March 2020. Four emergency declarations were subsequently issued between March 2020 and September 2021, during which the government advised people to refrain from going out and to avoid crowds. This may have affected children’s exercise habits.

Adequate physical activity is important for both physical and mental health [1,2]. Therefore, the World Health Organization (WHO) recommends moderate-to-vigorous physical activity (MVPA) for at least 60 min per day, at least 5 days per week [1]. However, according to an international collaborative study that collected data from 2011 to 2013, only 44.1% of preadolescent children met this recommendation [3]. In this context, multiple studies have reported decreased physical activity in children due to restrictions associated with the COVID-19 pandemic [4]. A systematic review by Kharel et al., 2022, found that of 34 studies worldwide that evaluated changes in physical activity in children and adolescents following the COVID-19 pandemic, 25 reported a decrease in physical activity [5].

On the other hand, a longitudinal study of 3–5 year-olds in 14 countries reported rather increased physical activity [6], as did our previous study of 6–7 year-olds [7]. As for trends by age, a U.S. study reported a more significant decrease in physical activity among preadolescent children (9–13 years old) than younger children (5–8 years old) [8]. A study conducted in Germany also found that lockdowns due to COVID-19 reduced the amount of time children spent playing sports, and that the reduction was greater in preadolescents and adolescents than in younger age groups [9]. Therefore, it is important to investigate how physical activity has changed in preadolescent children in Japan.

Changes in the physical activity of preadolescent children may have affected their physical function, such as muscle strength and balance. Muscle strength in preadolescence has been reported to predict muscle strength in middle adulthood [10]. Balance also develops during childhood [11,12]. In order to understand whether the activity restrictions which were imposed on children in relation to the COVID-19 pandemic could affect their future, it is important to assess differences in children’s physical function before and during the COVID-19 pandemic. The effects that COVID-19 restrictions have had on physical function has not yet been fully investigated in preadolescents in Japan.

We have been continuously assessing the physical function and lifestyle habits of healthy children in Japan since before the COVID-19 pandemic. In this study, we used the data to examine differences in physical activity, muscle strength, and balance among preadolescent children both before and during the COVID-19 pandemic.

## 2. Materials and Methods

### 2.1. Study Design and Study Population

This cross-sectional study was conducted in Okazaki City, in Aichi Prefecture in Japan. The population of the city is approximately 380,000, and 67,435 COVID-19 cases were reported by September 2022. Children aged 10 to 12 years who participated in the study were recruited from two of the forty-eight elementary schools in the city, and underwent physical function assessments between January 2018 and September 2022. We distributed flyers to all students of the appropriate age at the two schools, and invited them to participate. The assessments consisted of medical examinations by a pediatric neurologist and a pediatric orthopedic surgeon; questionnaires completed by the children and their parents; and measurements of body fat, grip strength, single-leg standing time, and a two-step test. In addition, participants were screened for intellectual disabilities using Raven’s Colored Progressive Matrices [13] and the Picture Vocabulary Test Revised (Nihon Bunka Kagakusha Co. Ltd., Tokyo, Japan).

Participants assessed from January 2018 to January 2020 were defined as the pre-COVID-19 group, and those assessed from January 2021 to September 2022 were defined as the during-COVID-19 group. The exclusion criteria included abnormalities of the nervous, respiratory, cardiovascular, ophthalmic, hematological, endocrine, or auditory systems; intellectual disability based on substandard scores of the Raven’s Colored Progressive Matrices and the Picture Vocabulary Test Revised; and missing data.

### 2.2. Data Collection

#### 2.2.1. Moderate-to-Vigorous Physical Activity Questionnaire

Questionnaires were administered to parents to determine the average number of hours of moderate-to-vigorous physical activity per week, excluding school physical education. The Japanese version of the World Health Organization Health Behavior in School-Aged Survey Questionnaire was used to assess MVPA levels [14].

#### 2.2.2. Body Fat

Body fat was measured using a multifrequency bioelectrical impedance analyzer (BIA; MC-780; Tanita, Tokyo, Japan). This measurement was performed more than 2 h after meals.

#### 2.2.3. Grip Strength

Grip strength was measured using an adjustable handheld dynamometer (GRIP-D; Takei Ltd., Niigata, Japan). The participants practiced once, and then performed the test with their dominant and nondominant hands. Grip strength was measured with the participants in a seated position, with their shoulder abducted and neutrally rotated, elbow in extension, and forearm and wrist in the neutral position [15]. The participants were instructed to grip the dynamometer handle as hard as possible and to hold it for seconds. The mean of the measurement results for both hands was used for evaluation.

#### 2.2.4. One-Leg Standing Time

One-leg standing time refers to the amount of time a participant can stand on one leg with their eyes open, and is representative of static balance function [16]. The participants were instructed to hold one foot off the floor and to keep it from touching the supporting leg. The time until the lifted foot touched the floor or supporting leg, or up to 120 s, was taken as the measurement value, and the maximum value for both legs was evaluated.

#### 2.2.5. Two-Step Test

The two-step test was conducted by measuring the maximum length of the two-step test that could be performed without losing balance, and dividing the measurement by body height [17]. Measurements were performed twice, and the maximum value was used in the analysis. The two-step test is representative of dynamic balance function [18].

### 2.3. Statistical Analysis

Paired difference tests were conducted to observe the differences in characteristics and physical function between the pre-COVID-19 and during-COVID-19 groups. The chi-square test was performed to test for differences according to sex. Based on the results of the Shapiro–Wilk and Levene tests, a two-sample *t*-test was applied for height and two-step test performance. The Mann–Whitney U test was used to compare the other variables.

Binomial logistic regression was then used to obtain the odds ratios of physical function with the pre- and during-COVID-19 groups as dependent variables (pre-COVID-19 = 0, during-COVID-19 = 1). The independent variables were grip strength as a measure of muscle strength, and two-step test value as a measure of balance function. Age and sex were included in the multivariable logistic regression model as adjustment variables. In this analysis, the possible values of the two-step test performance were too small to evaluate the odds ratio. Therefore, values multiplied by 100 were used.

Statistical analysis was performed using IBM SPSS Statistics version 28.0.1.0 (142) (IBM Corp., Armonk, NY, USA), and two-tailed *p* values < 0.05 were regarded as statistically significant.

## 3. Results

A total of 271 children, aged 10 to 12 years, were included in this study. In the pre-COVID-19 group, 16 children with underlying conditions (five post-trauma, four with Osgood’s disease, one with lumbar spondylolysis, one with knee arthritis, one who had undergone meniscectomy, one with severe back pain, one with amyotrophic lateral sclerosis, one under treatment for a pancreatic tumor, and one with an intellectual disability) and four children with incomplete data were excluded. In the post-COVID-19 group, five children were excluded (two after trauma, one with lumbar spondylolysis, one with plantar infection, and one under treatment for a knee tumor). A total of 177 children in the pre-COVID-19 group and 69 children in the during-COVID-19 group were included in the analysis. There were no duplicate participants in either group.

Table 1 shows the characteristics of each group. The during-COVID-19 group had a significantly higher body fat percentage and a significantly lower MVPA.

Table 2 summarizes physical functions by group. The during-COVID-19 group had weaker grip strength (*p* = 0.012) and performed significantly worse on the two-step test (*p* = 0.013). There was no significant difference in the one-leg standing time between the pre-COVID-19 and the during-COVID-19 groups (*p* = 0.749).

The results of the binomial logistic regression analysis of age, sex, and physical function are summarized in Table 3. The during-COVID-19 group was significantly related to weaker grip strength (*p* = 0.022) and worse performance in the two-step test (*p* = 0.028).

## 4. Discussion

The study found that MVPA in preadolescent children during the COVID-19 pandemic was less than that before the COVID-19 pandemic. In addition, the during-COVID-19 group had significantly weaker grip strength and worse performance in the two-step test. Logistic regression analysis revealed that the COVID-19 pandemic was significantly related to weaker grip strength and worse performance on the two-step test. To prevent adverse effects on children’s physical function caused by activity restrictions, it may be necessary to increase the time spent engaging in physical activity.

Grip strength is an indicator of total muscle strength [19]. Few reports have evaluated the effect of COVID-19 on muscle strength in children. A large cross-sectional study from Spain found that girls, but not boys, showed a trend toward reduced muscle strength during the COVID-19 pandemic [20]. In this study, logistic regression analysis showed an association between grip strength and the COVID-19 pandemic, even after adjusting for sex. The reason for this may be the difference in study period; the Spanish study included children from October 2020 to June 2021, immediately after the lockdown, whereas our study included children from January 2021 to September 2022, more than one year after the COVID-19 pandemic began, so our study results may reflect the effect of limiting physical activity over a longer period of time.

We evaluated dynamic balance using the two-step test. Dynamic balance is considered to be more accurate than static balance as a risk indicator. Physical activity has been reported to have positive effects on dynamic balance in children, adolescents, and older adults [21,22]. Dynamic balance is correlated with total body muscle strength [23,24], but logistic regression showed that two-step test performance was related to the COVID-19 pandemic independently of grip strength. This means that the loss of balance in preadolescent children caused by the activity restrictions associated with the COVID-19 pandemic may be due not only to total body muscle weakness, but also to a reduction in the opportunity to learn motor skills. It is, therefore, important to promote sports that promote a variety of motor skills, not limited to strength training.

We have already reported on the effect of the COVID-19 pandemic on physical activity and physical function of Japanese children aged 6 and 7 years. In children aged 6 and 7 years, the COVID-19 pandemic was associated with a slight increase in MVPA time and no significant difference in grip strength [7]. This contrasts with the decrease in MVPA time observed among the preadolescent children who participated in this study, which was similar to that reported in children in the United States and Germany after the COVID-19 pandemic [8,9]. Compared with younger children, preadolescents prefer more structured, social exercise [25]. This type of exercise may have been more likely to be limited by COVID-19-related restrictions that advised against going outside and visiting crowded places. In addition, many children in the upper school grades in Japan participate in club-based physical activities, which may have been restricted during the COVID-19 pandemic. In school-aged children, MVPA has been reported to be positively correlated with muscle strength [26,27]. Therefore, it is possible that reduced MVPA had a negative effect on muscle strength in preadolescent children in this study.

A longitudinal study by Martínez-Córcoles et al. [28] found that the COVID-19 pandemic had negative effects on static balance in school-aged children. In this study, the majority of participants in both groups had a maximum one-leg standing time value of 120 s; thus, this study did not find a significant difference in one-leg standing time in preadolescent children associated with the COVID-19 pandemic. Condon et al. [10] reported that one-leg standing time improved with age, especially after 7 to 8 years. Preadolescent children may be mature enough that static balance does not change if physical activity is limited. However, the ceiling effect should be considered in the interpretation of the results of one-leg standing time in this study.

Insufficient physical activity in children is associated with physical and mental health problems such as obesity, cardiovascular disease, and depression [1,2]. Low grip strength is associated with cardiometabolic risk [29,30]. Poor balance is a risk factor for injury from falls not only in older adults [1], but also in adolescents [31]. Therefore, it is important to monitor whether changes in physical activity and function associated with the COVID-19 pandemic are transient, or whether they have a permanent effect.

This study has several limitations. First, although the activity restrictions caused by the COVID-19 pandemic were global, this study was conducted in children from two elementary schools in one city. Therefore, the results may not be applicable to children living in other areas of the world. Second, this was a cross-sectional, not a longitudinal, study, so participation bias and generational effects cannot be excluded. Those who participated during the COVID-19 pandemic, at a time when most people tended to avoid going outside, may have had a higher awareness of the need for physical activity, or may have been more aware of a decline in their physical function. However, if a longitudinal study had been conducted on a single group of preadolescent children evaluated over time, the participants would have been affected by changes in their living environment and physical development as they grew older. Finally, it may also be useful to use other measurement items to assess physical function, such as 6-min walking distance, pulmonary function test, and stabilometry. For the evaluation of static balance in particular, it would be useful to combine one-leg standing with the center-of-gravity sway in future studies.

In conclusion, refraining from going outside and avoiding crowds due to the COVID-19 pandemic may have reduced physical activity in preadolescent children, which may have affected their muscle strength and balance. These results highlight the importance of ensuring that preadolescent children have adequate opportunities to participate in group exercise.

## Figures and Tables

**Table 1 healthcare-10-02553-t001:** Characteristics of the participants before and during the COVID-19 pandemic.

	Pre-COVID-19 (*n* = 177)	During-COVID-19 (*n* = 69)	*p* Value
Age (years) ^1^	11 (10–12)	11 (10–12)	0.480
Sex ^2^			0.095
Female	92	44	
Male	85	25	
Height (cm) ^3^	144.5 ± 8.2	146.3 ± 7.7	0.133
Weight (kg) ^1^	33.7 (22.9–74.4)	36.0 (25.4–77.9)	0.168
Body mass index (kg/m^2^) ^1^	16.4 (12.3–29.6)	16.6 (13.9–30.2)	0.582
Body fat (%) ^1^	13.6 (3.0–46.4)	15.6 (5.4–51.0)	0.016
MVPA per week (hour) ^1^	7.0 (0–24)	4.0 (0–17)	0.004

MVPA: moderate-to-vigorous physical activity. ^1^ The results are presented as the median (range), and groups were compared using the Mann–Whitney U test. ^2^ Groups were compared using the chi-square test. ^3^ The results are presented as the mean (standard deviation), and groups were compared using the two-sample *t*-test.

**Table 2 healthcare-10-02553-t002:** Physical function in participants, before and during the COVID-19 pandemic.

	Pre-COVID-19 (*n* = 177)	During-COVID-19 (*n* = 69)	*p* Value
Grip strength (kg) ^1^	15.8 (8.6–33.0)	14.4 (9.0–30.5)	0.012
One-leg standing time (s) ^1^	120 (29.0–120)	120 (11.9–120)	0.749
Two-step test ^2^	1.60 ± 0.14	1.56 ± 0.11	0.013

^1^ Data were analyzed using the Mann–Whitney U test, and the results are presented as the median (range). ^2^ Data were analyzed using the two-sample *t*-test, and results are presented as the mean (standard deviation).

**Table 3 healthcare-10-02553-t003:** Association between age, sex, physical function, and the COVID-19 pandemic.

	β	Odds Ratio ^1^ (95% CI)	*p* Value
Age (years)	0.344	1.411 (0.964–2.066)	0.077
Sex (female = 0, male = 1)	−0.409	0.664 (0.367–1.202)	0.177
Grip strength (kg)	−0.101	0.904 (0.829–0.986)	0.022
Two-step test ^2^	−0.025	0.976 (0.955–0.997)	0.028

^1^ Binomial logistic regression was used to obtain the odds ratios before and during the COVID-19 pandemic as dependent variables (pre-COVID-19 = 0, during-COVID-19 = 1). ^2^ The two-step test performance value was multiplied by 100. CI, confidence interval.

## Data Availability

The data presented in this study are available upon request from the corresponding author.

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
