# Peer review of "Physical Function of Japanese Preadolescents during the COVID-19 Pandemic"

_healthcare, 2022, doi:10.3390/healthcare10122553_

Round 1

Reviewer 1 Report

Thank you for your paper, an interesting analysis related to COVID 19 which has not been much researched so far.

Please complete the introduction part with relevant information about the age group you are discussing and their level of physical activity before COVID-19 from relevant international studies. Also, offer details of previous studies, even from other age groups of how COVID affected physical activity.

Reviewer 2 Report

The title is: Effect of the COVID-19 pandemic on physical function in 2 preadolescents in Japan. The authors write "This study 14 aimed to examine the changes in the physical function and physical activity of preadolescent 15 children before and during the COVID-19 pandemic".

In my opinion, both title and aims are inadequate for the text of this manuscript. We can read in the limitation that this was a cross-sectional, not a longitudinal study, so participation bias and generational effects cannot be excluded. I think that so kind of explanation is not enough.

It is a serious mistake when authors measure two different groups and conclude about something effect time. In addition, the groups have very different numbers of subjects. 

Are the authors sure that Covid affected? Maybe there are and were two different small groups of children. What kind of randomization was used? 

What was the theoretical idea to measure this parameter of physical fitness? What about endurance or trunk strength instead of one-leg standing time? What for?

There is lacking justification in the introduction for this study and therefore the discussion is weak too. The authors don't have somebody discuss or how explain the results.  Statistical methods are debatable.

In this analysis, the possible values of the two-step test performance were too small to evaluate the odds ratio. Therefore, the values multiplied by 100 were used" What it changed? 

Reviewer 3 Report

This manuscript represents the cross-sectional study with the aim to examine the changes in physical function and physical activity of preadolescent population before and during the COVID-19 pandemic. These are my comments and suggestions:

Abstract:

Abstract is clear and nicely written. It contains all the relevant elements.

Introduction:

Please, add paragraph on how worse performance in tests and lower levels of PA could negatively affect future health status.

Methods:

This chapter is adequate. If possible, comment on how did you estimate your sample size. Is 69 children in the during-COVID-19 group enough to detect difference?

Results:

No comments.

Discussion:

Comment clinical/public health relevance of your results. Why are they important? What are the possible consequences of worse performance and lower PA?

What would be your recommendation for future research?

Reviewer 4 Report

This is a well written paper presenting the impact of covid 19 on children’s strength. This evidence is important in understanding the consequences on well populations. 

Please confirm if Strobe guidelines submitted. 

Many thanks

Round 2

Reviewer 1 Report

Please add more details to the introduction part, describe all the aspects, related to age, COVID-19, physical activity which have been found in relevant international studies, also detail why this topic is important.

Author Response

Thank you for your suggestion. We add the following regarding age-dependent effects that the COVID-19 pandemic may have had on physical activity.

"A systematic review by Kharel et al. in 2022 found that of 34 studies worldwide that evaluated changes in physical activity in children and adolescents following the COVID-19 pandemic, 25 reported a decrease in physical activity [5].

On the other hand, a longitudinal study of 3-5 year olds in 14 countries reported rather increased physical activity [6], as did our previous study of 6-7 year olds [7]." (page 1, lines 42-47)

We have also detailed the age of the subjects in the citation of one study to clarify the age of preadolescence we are discussing.

"As for trends by age, a U.S. study reported a decrease in physical activity among preadolescent children (9-13 years old) than younger children (5-8 years old) [8]. " (page 2, lines 47-49)

We added that our study is significant as a report from Japan, as many countries have already reported age-dependent changes in physical activity associated with the COVID-19 epidemic.

"Therefore, it is important to investigate how physical activity has changed in preadolescent children in Japan." (page 2, 52-53)

We have described more clearly the significance of the changes in the preadolescent's physical functions, which was the point we most wanted to evaluate in this study.

"In order to understand whether the activity restrictions we have imposed on children in relation to the COVID-19 pandemic could affect their future, it is important to assess differences in children's physical function before and during the COVID-19 pandemic." (page 2, 57-60)

Reviewer 2 Report

 Is very good that the authors changed the title.  Thank you for every modification.  But in my opinion, there are too less subjects to consider anything.

Author Response

Thank you for giving us the opportunity to strengthen our manuscript with your valuable comments and queries.

You have raised an important point. Post hoc power calculations using G*Power (Heinrich Heine University of Düsseldorf, Düsseldorf, Germany) for MVPA (7.7±5.5 vs. 5.5±4.9), grip strength (16.2±4.0 vs. 15.1±4.2) and two-step test values (1.60±0.14 vs. 1.56±0.11) were 0.82, 0.46 and 0.61 respectively. We would like to conduct future research with a larger number of subjects.